# Changes in the Composition of the Soil Bacterial Community in Heavy Metal-Contaminated Farmland

**DOI:** 10.3390/ijerph18168661

**Published:** 2021-08-16

**Authors:** Shu-chun Tseng, Chih-ming Liang, Taipau Chia, Shan-shin Ton

**Affiliations:** 1Department of Environmental Engineering and Science, Feng Chia University, Taichung City 407802, Taiwan; sctseng0315@gmail.com (S.-c.T.); cmliang@fcu.edu.tw (C.-m.L.); 2Department of Safety, Health and Environmental Engineering, Hung Kuang University, No. 1018, Sec. 6, Taiwan Boulevard, Shalu District, Taichung City 433304, Taiwan; tpchia38@gmail.com

**Keywords:** heavy metals, soil bacteria, bacterial community

## Abstract

The structural changes of microorganisms in soil are the focus of soil indicators research. The purpose of this study was to investigate the changes in the composition of the soil bacterial community in heavy metal-contaminated soil. A total of six soil samples (two sampling times) were collected from contaminated farmland at three different depths (surface, middle, and deep layer). The pH value was measured. The concentrations of heavy metals (Cr, Ni, Cu, Zn, Cd, and Pb) and the soil bacterial community were analyzed using ICP-OES and 16S rRNA gene sequencing. The results of the two samplings showed that the pH value in the deep layer decreased from 6.88 to 6.23, and the concentrations of Cu, Zn, Cd, and Pb, with a smaller ion radius, increased by 16–28%, and Shannon, Chao1 increased by ~13%. The bacteria community composition at the three depths changed, but Proteobacteria, Acidobacteria, and Actinobacteria were the dominant phyla. In the copper and zinc tolerance test, the isolated bacterium that was able to tolerate copper and zinc was *Bacillus* sp. We found that, the longer the heavy metal pollution was of concern, the higher the tolerance. These results can be used as references for the microbial remediation of heavy metal-contaminated farmland.

## 1. Introduction

Soil is an important component of the natural ecological environment, and it is of great value in maintaining plant productivity and supporting human survival [1]. Microorganisms are key players in many soil functions, such as biogeochemical cycling, plant productivity, and climate regulation, and are essential for the integrity of terrestrial ecosystems [2]. Soils also provide complex and diverse habitats for microorganisms, which are spatially and temporally heterogeneous in their physical, chemical, and biological properties [3]. Distinct soil environments may range from a few micrometers to millimeters apart, and their microbial abundances, rate of microbial activity, abiotic characteristics, and composition of the microbial community may differ [4].

Global environmental degradation has increased dramatically due to anthropogenic activities, particularly the extensive use of heavy metal-bearing chemicals in agriculture, transportation, and the chemical industry [5]. Heavy metals exist in various forms, such as free metal ions, interchangeable metal ions, soluble metal complexes, and metals bound in other compounds. These different forms result in different levels of bioavailability and toxicity in agricultural soils, and their mobility is also influenced by different factors [6]. Over the past few decades, environmental contamination by heavy metals has become an extremely important issue, and can affect the water, air, and soil, creating a variable and irreversible cycle of toxicity [7]. Soil is a rich habitat, containing all major groups of microorganisms. However, significant levels of heavy metal accumulation can change the soil composition, mineral recycling, and associated metabolic activities, and exert a selective pressure on soil microbiota. Thereby, heavy metal contamination can affect the functioning of microorganisms, leading to morphological and physiological changes in microbial population structures. Ultimately, it may decrease microbial biomass and activity, as well as affect the structure and diversity of soil microbial communities [8,9].

Microbes have evolved resistance to heavy metal stress. This can be attributed to the fact that they are able to mobilize, sequester, or transform various ions, thereby affecting the metals’ biogeochemical mobility [10]. Although heavy metals could have a detrimental effect on some microbial populations, their selection pressure can stimulate the growth of tolerant microbes, leading to changes in microbial community diversity [11]. In Li’s study [12], their results also supported that soil microbes adapt to long-term heavy metal pollution through a change in microbial community composition and structure, rather than a change in their diversity and evenness. However, most studies have focused on the effects of heavy metals on root exudations, soil microbial activity, soil enzyme activity, and so on. There is little information available about how heavy metals regulate the entire microbial community [13]. This information, alongside changes in the composition of microorganisms and the growth of tolerant microbes in response to heavy metal toxicity, are important biological indicators.

To ensure the sustainable use of soil and groundwater resources and the protection of the health of the public, the EPA (Environmental Protection Agency) in Taiwan has undertaken efforts in the pollution investigation of farmland, factories, industrial parks, and other potentially polluted sites. The “soil or groundwater pollution control site”, defined by the Taiwan EPA, is a site at which the metal concentration in the soil exceeds the control standard. At the end of 2018, a total of 7253 contaminated farmland sites (around 1138.5 hectares) across the country had been announced as heavy metal pollution control sites [14]. There are 607.1 hectares of control sites in Changhua County, which is ~53% of the total controlled sites in Taiwan. The farmland in Changhua County has a severe contamination problem. Taiwan’s land is very limited and densely populated, and there is limited arable farmland. For heavy metal-contaminated soil, the most used remediation is the soil dilution method, which has failed to solve the problem. Therefore, studying the changes in the microbial community structure in heavy metal-contaminated soil will help soil remediation efforts, using biological treatments.

The objective of this study was to explore whether the composition and the structure of microbial communities vary with the depth of the soil and the concentrations of metals in heavy metal-contaminated farmland. The bacterial species that can effectively adsorb and tolerate the heavy metals copper and zinc were also analyzed.

## 2. Materials and Methods

### 2.1. Soil Sampling

The experimental soils were obtained from a rice paddy field, shown in Figure 1 (24°04′53.3′′ N, 120°28′34.5′′ E). The abandoned farmland in this study is a long-term heavy metal (mainly Cu and Zn)-contaminated site. Point A_0_ was the sampling site for the pollution survey by the Taiwan EPA. In order to obtain the average contamination concentration, the sample was a mixture of the contaminated soil from the control site. Point A_1_ is the hotspot of the contaminated farmland, the sampling site in this study, and is close to the irrigation inlet. The irrigation water containing heavy metals (mainly Cu and Zn) from industrial wastewater led to the accumulation of heavy metals in the agricultural soil, which exceeded the concentrations of the regulated criteria. The study area was announced as a pollution control site in 2019, and farming was prohibited. No further soil treatment occurred before our study. The grab sample method was chosen in this study, the period between the two samplings was eight months, and the samples were collected in the same location on 3 February 2020 (sampling I) and 14 October 2020 (sampling II). The rainfall period in this area is concentrated between May and September, as shown in Figure 2. There were two observation records of the monthly cumulative rainfall exceeding 200 mm in May and August. The changes in heavy metal concentrations and the structure of the bacterial community under natural conditions were investigated and compared.

Soil samples were collected at the highly polluted hotspot from three soil depths: the surface soil (SL_I_, SL_II_, depth of 0–10 cm), middle soil (ML_I_, ML_II_, depth of 10–15 cm), and deep soil (DL_I_, DL_II_, depth more than 15 cm). All the collected soil samples were packed in sealed clean bags. They were stored at low temperatures until they were sent back to the laboratory. The samples were divided into two groups; the first of the soil samples was stored at −80 °C for DNA extraction and high-throughput microbial sequencing. The other soil samples were air-dried and screened through a 2 mm screen to remove stones, plant roots, and other non-soil materials. Then, the soil samples were homogenized and stored for further physical and chemical analysis.

### 2.2. Analysis of Heavy Metals in Soil

In this study, the aqua-regia digestion method was used to analyze the total concentration of heavy metals in the soil samples. All the samples were analyzed in triplicate, and 0.5 g of the soil sample was mixed with concentrated nitric acid and concentrated hydrochloric acid, and then subjected to microwave digestion and extraction. The heavy metal concentrations of the extracted solution were quantified and analyzed using the Inductively Coupled Plasma Spectrometer (ICP-OES, Agilent 5110 series, Agilent Technologies Taiwan Ltd., Taipei City, Taiwan).

### 2.3. DNA Extraction, PCR Amplification, and Sequencing

Total genomic DNA from the soil samples was extracted using the CTAB/SDS method [15]. Amplicons of the polymerase chain reaction on the 16S rRNA gene was used for microbial diversity analysis. For the PCR reaction, total genomic DNA was used as the template and the specific primer set 341F (5′-CCTACGGGAGGCAGCAG-3′) and 518R (5′-ATTACCGCGGCTGCTGG-3′) was used for amplifying the 16S rRNA gene [16]. Next, 25 μL of PCR amplification was performed, and the mixture contained 0.5 μL of KAPA© High-Fidelity PCR Master Mix (KAPA BIOSYSTEMS, Bath, UK), 0.5 μM of forward and reverse primers, and about 1 ng of template DNA. Thermal cycling started with the initial denaturation at 95 °C for 3 min, followed by 30 cycles of denaturation at 95 °C for 30 s, annealing at 57 °C for 30 s, and elongation at 72 °C for 30 s. Finally, the samples were kept at 72 °C for 5 min. Mixing equal volumes of 1× loading buffer (contained SYB green) with the PCR products, electrophoresis was carried out on 2% agarose gel for detection. Samples with one bright main strip between 450 and 500 bp were chosen for further experiments. PCR products were mixed in equidensity ratios. Then, the mixed PCR products were purified with a QIAquick Gel Extraction Kit (QIAGEN, Hilden, Germany). Sequencing libraries were generated using the Truseq nano DNA Library Prep Kit (Illumina, San Diego, CA, USA) following the manufacturer’s recommendations. The library quality was assessed on the Qubit@ 2.0 Fluorometer (Thermo Scientific, Waltham, MA, USA) and Agilent Bioanalyzer 2100 system. The initial data directly generated by the sequencer were the original data. We used Trimmomatic (v0.39) [17] to remove the original data from the adapter sequence to obtain clean data. Then, we removed the additional primer sequence, spliced the two reads (R1, R2), undertook chimera removal, and obtained the final effective data. Lastly, the library was sequenced on an Illumina Miseq platform, generating 300 bp paired-end reads. Sequences were clustered into OTUs (operational taxonomic units) at 97% similarity, and the most abundant sequence was selected from each OTU as the representative of the respective OTU [18]. The OUT sequences were compared with the NCBI database to confirm the information of each OUT species.

### 2.4. Diversity Analyses

The OTUs obtained after analysis were used for the re-estimation of the microbial species diversity in the sample. The richness analysis for a single sample was conducted using QIIME (Quantitative Insights into Microbial Ecology). Chao1 uses the number of species that only appear once or twice to perform the calculation weighting, mainly emphasizing the existence of rare OTUs [19]. A higher index value is indicative of a greater richness in the community. The Shannon diversity index is used to estimate the diversity of clusters; the higher the index value, the greater the diversity. Simpson describes the probability that two species drawn from the population belong to the same species. The higher the value, the lower the diversity, while in the Simpson_reciprocal, the higher the value, the higher the diversity [20].

### 2.5. Domestication, Separation and Purification of Heavy Metal-Tolerant Bacteria

In this step, 30 g of each soil sample was mixed with 90 mL of heavy metal-free nutrient broth medium (NB). After shaking for 30 min, the evenly shaken soil mixture stood was left for 30 min to allow the large particles of soil to settle by gravity. Then, 10 mL of the supernatant liquid was taken and added into a 250 mL Erlenmeyer flask containing different copper and zinc concentrations (50 mg/L, 100 mg/L, 200 mg/L) in nutrient broth, and the pH was adjusted to 7 with 1 N NaOH/1 N HCl. The Erlenmeyer flask was kept at 30 °C, shaken at 120 rpm, and cultured for 8 h, before being diluted 10–1000 times. Subsequently, 100 μL of cell suspensions was taken from each dilution and dropped on nutrient agar. A triangular glass rod was used to evenly smear the water sample on the solid medium to a non-flowing state, and the samples were then inverted and incubated at 37 °C for 48 h. The colonies were determined visually, and single colonies of different types were selected for purification.

### 2.6. Sequencing and Comparison of Bacterial Species

The DNA of purified colonies were extracted using a Total DNA Extraction Kit (product number: atci-DNA, Taco), by a taco automatic nucleic acid extractor (magnetic bead system). After DNA extraction, 16S rRNA PCR amplification of the purified and separated bacteria was carried out with 16S rRNA. As the PCR was not amplified in a way that is easily attributable to bacterial diversity, three different sets of universal primers (27F/1525R, 8F2/806R, and fD1modF/16S1RR-B) were adopted for the amplification of 16S rRNA, as shown in Table 1. Sequencing and decoding were conducted for successfully amplified and longer PCR products. NCBI’s Nucleotide BLAST function was used to perform bacterial comparison and sequencing in the DNA database.

### 2.7. Heavy Metal Tolerance Test of Isolated Bacteria

In order to evaluate the tolerance of purified bacteria to heavy metals, the isolated bacteria were cultured in nutrient agar with different copper and zinc concentrations (50, 100, 200, 400, and 600 mg/L). In addition, nutrient agar without heavy metals was used as the control, after inversion and culturing at 37 °C for 48 h. The microorganisms with higher tolerance in the soil were isolated in the heavy metal copper and zinc culture solution. The purified and separated pure bacteria were cultured on a solid medium containing different heavy metal concentrations, and the growth of the microorganisms was investigated.

## 3. Results and Discussion

### 3.1. Soil Element Content and Properties

In order to understand the effects of heavy metal concentrations on the bacterial community in contaminated farmland after a period of time, six soil samples were collected at a highly polluted hotspot from different depths. The heavy metal concentrations at three different depths for the two samplings are shown in Table 2. Since the sampling location was a hotspot of the polluted site, the concentration values in this study are much higher than the monitoring standards of regulation, which were determined from the composed sampling. Among them, the Cu concentrations were recorded between 460.13 and 557.09 mg/kg, which were much higher than the soil control standard values (food crop agricultural control standard, 200 mg/kg) [24,25]. The Zn concentrations, between 544.8 and 603.06 mg/kg, were above the soil monitoring standard, and close to the control standard value (food crop agricultural control standard, 600 mg/kg). The concentrations of Cr, Cd, and Pb in the six samples were all lower than the monitoring standard. The concentration of Ni in the first sampling exceeded the monitoring standard, but it decreased at the second sampling, which was lower than the monitoring standard (SL_II_ and ML_II_) and the control standard (DL_II_).

After eight months without any treatment, the concentrations of heavy metals at different depths changed in the second sampling. The variation of the concentrations of heavy metals in the SL, ML, and DL are shown in Table 2. Compared to the first sampling, the concentrations of Cu, Zn, Cd, and Pb in DL_II_ all increased, while the concentrations of Cr and Ni decreased. The reason for the change may be that all the pH values of the soil decreased after the first sampling. The cation exchange capacity decreases with increasing acidity, and even the surface charge of soil changes from negative to positive. The attraction and repulsion of cations to the charged colloid surface occurs according to Coulomb’s law. The exchange selectivity of cations is determined not only by their valence, but also by the relative hydrated size of the same valence [26]. Therefore, a heavy metal with a smaller ionic radius will increase its mobility in the soil solution, due to the positive charges repelling one another. This means that the Cu, Zn, Cd, and Pb ions with the smaller radius have higher mobility in acidic soil, while Cr and Ni ions with a larger radius and less mobility underwent a smaller change in the DL. It has also been mentioned by Khadhar et al. that the change of pH promotes the transformation of Pb, Cu, and Zn from the most residual fraction to the more mobile, exchangeable, and reducible fractions, which impacts metal distribution [27].

Before the second sampling, the farmland experienced a daily cumulative rainfall of over 50 mm for two days, and a monthly cumulative rainfall of over 200 mm for two months during rainy season. As a result, the soil acidity at different depths had decreased. Under a weak acidic soil environment, the metal ions dissolved in the soil solution, which changed their distribution in the soil through horizontal diffusion and percolation downward. The pH of each layer increased slightly with increasing depth, which also helped with the recapturing of cations in the deep layer soil. Table 2 shows that the changes of the concentrations of Cr and Ni with a larger ion radius were slightly decreased at DL_II_, whereas the changes of the concentrations of Cu, Pb, Zn, and Cd with a smaller ion radius increased by over 16%. This means that heavy metal ions with a smaller radius moved to the deeper soil layers through percolation.

Ayangbenro indicated that the solubility and bioavailability of heavy metals can be influenced by a small change in the pH level. At acidic pH levels, heavy metals tend to form free ionic species, with more protons available to saturate metal-binding sites [28]. This may explain the concentration changes in our study with lower pH values in the second sampling. However, another concern is that, at a lower pH, many heavy metals (including Cu, Zn, and Pb) may combine with organic matter to form solid substances in the soil, so that the heavy metals become fixed in the soil [29]. This should be considered in future studies.

### 3.2. Sequence Data and Bacterial Taxonomic Richness

From the six soil samples collected from the two samplings, a total of 550,979 high-quality 16S rRNA gene reads were obtained after Illumina Miseq sequencing, after eliminating the wrong sequences. An effective sequence was classified by a similarity of 97%, and there was a total of 158,167 OTUs from these readings. The rarefaction curve is shown in Figure 3. The differences in the OTUs at different depths at the same sampling times were not high, as shown in Table 3. In the first sampling, the OTUs from the three depths were 33,130, 39,622, and 30,014, respectively, and they were 21,681, 18,085, and 15,635 at the second sampling, respectively. After leaching by rainfall, all OTUs at the three depths decreased in the second sampling. As shown in Figure 4 (the Venn diagram), in the first sampling, there were 196 core OTUs across the three different depths. SL_I_, ML_I_, and DL_I_ had unique OTU values of 81, 63, and 33, respectively. In the second sampling, there were 207 core OTUs across the three different depths. SL_II_, ML_II_, and DL_II_ had OTU values of 30, 59, and 48, respectively. As the soil was polluted by heavy metals for a long time without treatment, the concentrations of heavy metals in SL and ML remained high, which resulted in a significant decrease in the OTU value at the second sampling. This result is consistent with Thavamani’s finding that soil microorganisms and enzyme activity were reduced with higher levels of mixed contamination [30]. In addition, Cd, Cu, and Zn have been reported to decrease microbial biomass in similar studies [31]. Under long-term heavy metal pollution, the diversity and abundance of the soil microbial community decreased [32].

In this study, Chao1 was used to estimate the richness, and Shannon was used to estimate the species diversity. Table 3 shows the analysis for the soil bacterial richness and diversity. Comparing the two samplings, the change of Chao1, Shannon, and Simpson_reciprocal had similar trends at three different depths. In the second sampling, the Chao1, Shannon, and Simpson_reciprocal values in SL_II_ and ML_II_ were all reduced. On the contrary, the Chao1, Shannon, and Simpson_reciprocal values in DL_II_ were all increased by ~12%. When the untreated farmland was sampled for a second time after 8 months, the concentration of heavy metals in DL_II_ had increased (Table 2), and the abundance and diversity of the microbial community had also increased (Table 3). Studies by Lee et al. [33] and Hu et al. [34] also showed that the microbial community structure might vary greatly at different soil depths. The microbial community structure in soil polluted by heavy metals significantly changed with depth [35]. Their results are consistent with our study, in that Cu, Zn, Pb, and Cd with a smaller ion radius moved to deeper soil layers through percolation in a weakly acidic environment (Table 2), resulting in higher metal concentrations (15.8–28.0%) in DL_II_, and higher Chao1 and Shannon values in DL_II_ (~12.8%). This positive trend was not observed for Cr and Ni. It is suggested that, due to the property of microbial oligodynamic action of the heavy metal, the amount and type of the heavy metal could be the factor that affects the microbial diversity [36].

In a further analysis of the relationship between the metal concentrations with Chao1, Shannon, and pH, Figure 5 shows that the pH was significantly and negatively correlated with the Cu/Zn/Pb concentration (smaller ion radius), but significantly and positively correlated with the Cr/Ni concentration (larger ion radius) (*n* = 6). This correlation analysis demonstrated both the Chao1 and Shannon have weak negative correlations with pH value. Li discusses that some bacterial communities were highly resistant to the lower pH (pH < 6.5) or more abundant in alkaline soil (pH > 7.5) [37]. The pH seems not to be a major impact factor for microbial composition change. However, in our study, even the change of pH was slightly reduced from 6.88 to 6.09. The decreasing pH promotes the mobility of Cu, Zn, Cd, and Pb, and subsequently enhanced the diversity of soil microbial. There was no obvious relationship between the Cr/Ni concentrations and the Chao1 or Shannon. Chao1 was less strongly correlated with the pH value and all the metal concentrations. Further studies with more samples are suggested to confirm these relationships.

### 3.3. Analysis of the Bacterial Community in Soil

After sequencing and analysis, there were 4 phyla of archaea and 26 phyla of bacteria found in the six soil samples. Figure 6 shows the change in the abundance of the bacterial community composition in different samples. The dominant bacteria phyla in SL_I_, ML_I_, and DL_I_ were Proteobacteria (52.29%, 42.45%, and 55.33%, respectively), followed by Chloroflexi (21.68%, 28.40%, and 21.05%, respectively), and Acidobacteria (11.39%, 10.60%, and 9.80%, respectively). In SL_II_, ML_II_, and DL_II,_ Proteobacteria, Chloroflexi, and Acidobacteria were the dominant bacteria phyla_._ However, the relative abundance changed to Proteobateria (21.52%, 25.79%, and 31.32%, respectively), followed by Chloroflexi (31.21%, 30.56%, and 28.84%, respectively), and Acidobacteria (22.32%, 21.55%, and 18.43%, respectively). Proteobacteria all decreased across the three depths, while Chloroflexi and Acidobacteria increased. The dominant phyla that were observed in the paddy soil samples included Proteobacteria, Chloroflexi, Acidobacteria, Actinobacteria, Gemmatimonadetes, Verrucomicrobia, Thaumarchaeota, Firmicutes, and Nitrospirae [38]. These predominant phyla were also obtained in other metal-contaminated soil, such as paddy soil, mining sites, and sediment, which indicates that these phyla might be closely related to metal-contaminated soil [39]. Overall, the community composition analyses indicated that variation in community structure is present, with metals, pH, and soil moisture explaining a significant proportion of the community variation. This result is supported by 16S rRNA gene sequencing, indicating that these metals also correlate to significant increases in OTUs within the phyla Acidobacteria, Actinobacteria, Bacteroidetes, Chloroflexi, Planctomycetes, Proteobacteria, and Verrucomicrobia [40].

In addition, the bacteria phyla with the higher relative abundance of the two samplings was investigated (in Figure 6). It can be seen that the relative abundance of Proteobacteria across the three depths reduced, while the relative abundance of other dominant bacteria of SL_II_ increased. The relative abundance of Proteobacteria decreased in ML_II_, and Actinobacteria, Crenarchaeota, and Patescibacteria also decreased slightly. Only Proteobacteria and Patescibacteria reduced in DL_II_, and all other dominant phyla increased. It can be seen that changes in the concentration of heavy metals led to changes in the microbial flora in the soil.

The dominant bacteria phyla varied with the metal pollution at different depths. Li [12] indicated that the responses of the phyla Proteobacteria to heavy metals varied among studies; both significant positive and negative correlations with heavy metals were found. The variation of the response of Proteobacteria to heavy metals may due to the phylum exhibiting a complex lifestyle, and being able to use various forms of organic matter as carbon, nitrogen, and energy sources [41].

Regarding the genera changes in the two samplings, the number of genera decreased from 300 to 223 in SL and from 289 to 266 in ML, and increased from 228 to 257 in DL. In the first sampling, *Chlorflexi*, *Sulfurifustis,* and *Thiobacillus* existed at all three depths. In the second sampling, a high proportion (more than 10%) of *Chlorflexi* still existed at all three depths. *Sulfurifustis* was no longer the dominant genus in SL_II_, and it was also reduced in ML_II_ and DL_II_*. Anaerolineae*, *Holophaga_*sp., and *Bryobacter* were the dominant bacteria genera at the three depths, as shown in Figure 7. However, the heavy metal tolerance of *Anaerolineae*, *Holophaga_*sp., and *Bryobacter* found in our study has rarely been mentioned in other heavy metal-related studies. Nunoura [42] noted the relatively high abundance of the *Anaerolineae* species in a microbial ecosystem in which hydrogenotrophic organisms were scarce in the population. Previously, a relatively high abundance of *Anaerolineae* phylotype strains in man-made environments has been observed under methanogenic conditions, and some of the *Anaerolineae* strains from such environments have been obtained from syntrophic enrichment with methanogens. It has been shown that rhizosphere bacteria may increase the heavy metal concentrations in hyperaccumulator plants significantly. In the rhizosphere, a high percentage of bacteria belonging to the *Holophaga/Acidobacterium* division and α-*Proteobacteria* were found [43].

### 3.4. Bacteria Identification

A dilution series of the two soil samplings in 200 mg/L copper and zinc heavy metal broth was prepared. In the heavy metal-tolerant culture, two pure bacteria tolerant to copper (200 mg/L) were selected from the first sampling. In the second sampling, 10 strains of copper-tolerant pure bacteria and four strains of zinc-tolerant pure bacteria were selected. Using 16S rRNA sequencing, the main strains were *Bacillus* sp., and some strains, namely *Bacillus velezensis,* coexisted in SL_II_, ML_II_, and DL_II_. *Bacillus marisflavi* existed in both SL_II_ and DL_II_; *Bacillus albus* tolerated domestication in both copper and zinc. Although *Bacillus albus* was isolated from both the first sampling and the second sampling, it was isolated only from the copper broth in the first sampling, and it was isolated only from the zinc broth in the second sampling. All existed in SL, ML, and DL. In the second sampling, there were a few strains able to tolerate zinc, including *Bacillus albus* (Table 4). The WHO has reported that, among the genera, many members of *Bacillus* have been found to be useful, and have been isolated for use in microbial remediation or phytoremediation for Cu, Cd, Zn, and Pb [6]. As we found in this study, *Bacillus albus* is a strain that can tolerate domestication in both copper and zinc. Currently, biopreparations based on *Bacillus* sp. bacteria are very popular, because they proliferate rapidly. *Bacillus* sp. strains have received a GRAS (Generally Recognized as Safe) designation from the United States Food and Drug Administration to prove their lack of toxicity or pathogenicity [44]. Thus, it is speculated that the *Bacillus* sp. found in our study might be useful for microbial remediation.

### 3.5. Tolerance to Cu and Zn

The microorganisms with higher tolerance in the soil were isolated from a specific liquid medium containing copper and zinc, and the tolerance growth of each microorganism was observed in the copper and zinc medium. The results are shown in Table 4. In the copper-containing medium, the coding numbers with Cu_SL_II_-1, Cu_SL_II_-2, Cu_ML_II_-2, Cu_ML_II_-4, and Cu_DL_II_-3 showed the best tolerance. These can grow in a medium containing 200 mg/L copper, but exceeded their tolerance limit beyond 400 mg/L. In the zinc medium, the coding numbers with Cu_SL_II_-2, Cu_ML_II_-3, Zn_SL_II_-1, Zn_SL_II_-2, and Zn_ML_II_-1 demonstrated the best tolerance, and they grew well on the medium containing 200 mg/L zinc. In Cu_ML_II_-2, Cu_DL_II_-3, and Zn_DL_II_-1, the microorganism growth can still be observed in the medium containing 200 mg/L zinc. Again, beyond 400 mg/L, they exceeded their tolerance limit.

In the investigation of the tolerant strains in the copper- and zinc-containing medium, the tolerance growth of the pure bacteria was observed in the solid medium with different concentrations of copper and zinc. The coding numbers with Cu_SL_II_-2, Cu_ML_II_-2, and Cu_DL_II_-3 were found to be the best strains at high concentrations of copper and zinc (200 mg/L). The copper-tolerant strains from the soil samples that were isolated from the copper-containing solid medium could also tolerate the zinc-containing liquid medium, but the zinc-tolerant strains from the soil samples isolated from the zinc-containing solid medium culture solution were not able to grow well in the copper-containing liquid medium (Table 4). Bacterial communities can exhibit structural and functional resilience to metals. Heavy metal pollution inhibits microbial activities, affects bacterial community structure, and induces further bacterial community tolerance to heavy metals [45]. Zhu et al. [46] suggested that these properties protect other species and increase the abundance of rare species. In our study, the isolated bacteria that best tolerated copper and zinc was *Bacillus* sp. We found that the longer the heavy metal pollution time, the higher the tolerance. *B. pumilus* and *B. cereus* isolated from the soil samples were found to be resistant to several heavy metals in Colak’s study [43]. They demonstrated higher biosorption capacities in Cu(II) and Zn(II) solution. Other studies have shown that the isolated *Bacillus cereus* RC-1 is resistant to Cd(II) [47], and *Bacillus* sp. is resistant to Pb(II) from lead–zinc mine soil [48]. The resistance pattern of *Bacillus cereus* NWUAB01 isolated from mining soil showed that the organism tolerated Pb better than Cd and Cr [49]. In summary, *Bacillus* sp. can be isolated from heavy metal-contaminated soils or mining areas, not only for their resistance, but also for the removal of soil contaminated by Mn [50], Pb [51], Cd [44,46], and Cr [46].

## 4. Conclusions

The changes in the composition and structure of the soil bacterial community at different depths in a hotspot of a farmland contaminated by heavy metals were studied. In the first sampling, the soil had lower abundance and diversity in DL. The dominant bacteria phyla were Proteobacteria, Chloroflexi, and Acidobacteria. However, the farmland experienced heavy rainfall between the two samplings. In the second sampling, the dominant bacteria remained the same, but Proteobacteria, which were less resistant to heavy metals, reduced by about 25%. The OTUs and diversity in SL and ML decreased. While most of the metal concentrations in deep soil increased at the second sampling with a lower pH, the abundance and diversity of the soil bacterial communities increased. We found that the slightly reduced pH promoted the mobility of Cu, Zn, Cd, and Pb and subsequently enhanced the diversity of the soil microbial community. However, further studies with more samples are needed to investigate other influences on the composition and structure of the soil bacterial community, such as the organic matter, total carbon, total nitrogen, and heavy metal interactions.

Our results showed that *Anaerolineae*, *Holophaga*_sp., and *Bryobacter* existed at all three depths of the second sampling and were tolerant to heavy metals, which were rarely mentioned in other heavy metal-related studies. In the two samplings, two strains of copper and zinc were isolated from the first sampling, and 10 strains were isolated from the second sampling, mainly of *Bacillus* sp. The bacterial communities in the soil developed adaptability and resistance to heavy metal pollution over a long period of time, and this finding could be used as a reference for the microbial remediation for heavy metal-contaminated farmland in the future.

## Figures and Tables

**Figure 1 ijerph-18-08661-f001:**
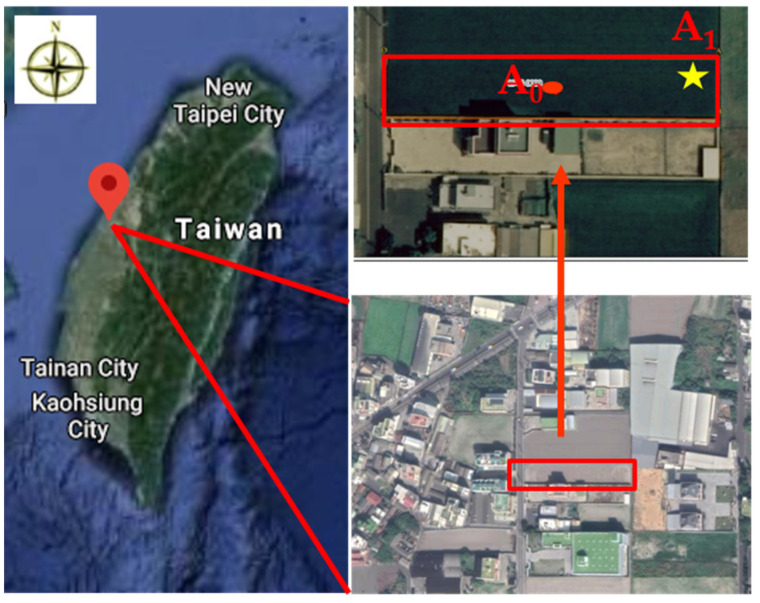
Study area and sampling site in Changhua County (A_0_; EPA survey site; A_1_; hot spot/sampling site).

**Figure 2 ijerph-18-08661-f002:**
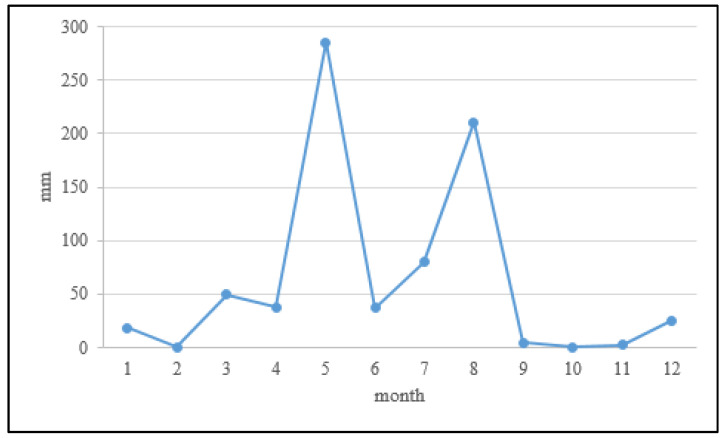
Monthly cumulative rainfall statistics chart during the sampling period.

**Figure 3 ijerph-18-08661-f003:**
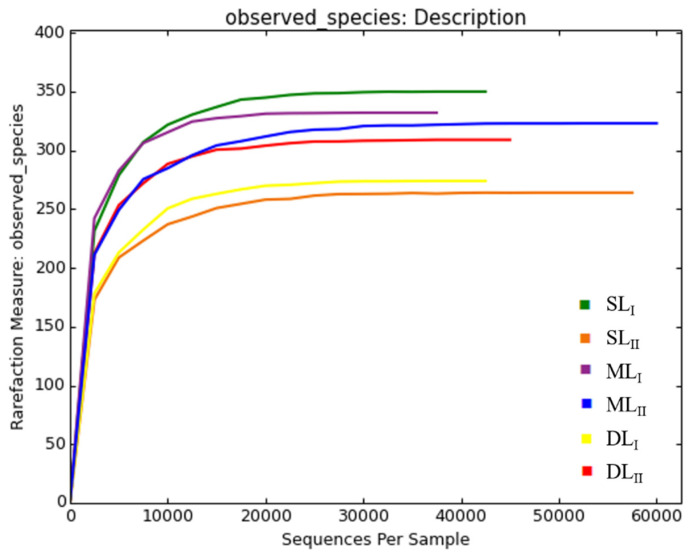
Rarefaction curve of the observed bacterial species in the different soil samples.

**Figure 4 ijerph-18-08661-f004:**
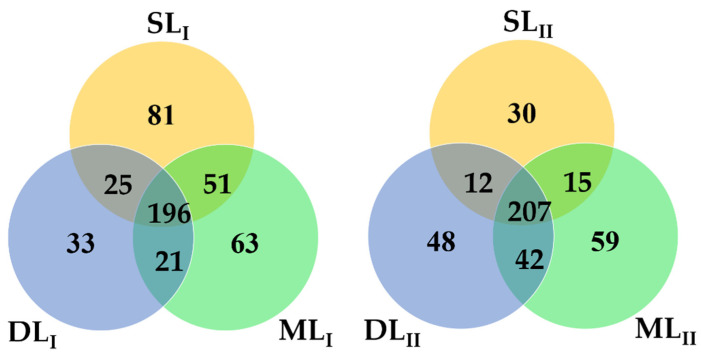
Venn diagram showing differences in the composition and structure of microbial OTUs under different soil heavy metal contamination levels.

**Figure 5 ijerph-18-08661-f005:**
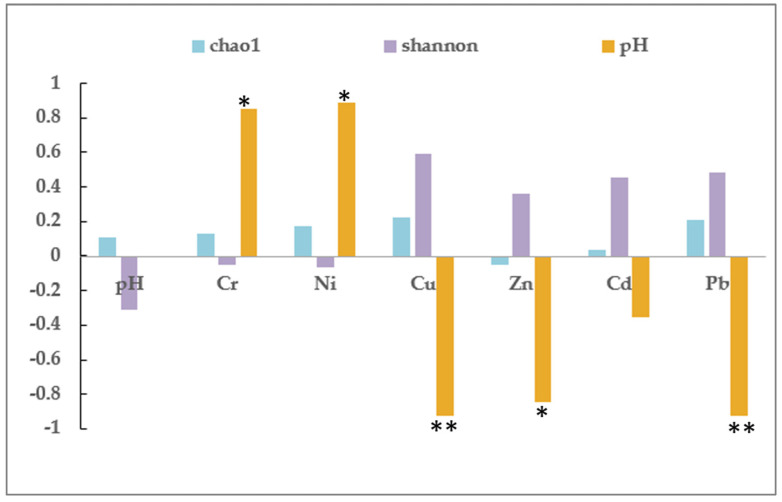
Correlation coefficients of the heavy metal concentrations and the Chao1, Shannon, and pH values (Pearson test; *: *p* < 0.05; **: *p* < 0.01).

**Figure 6 ijerph-18-08661-f006:**
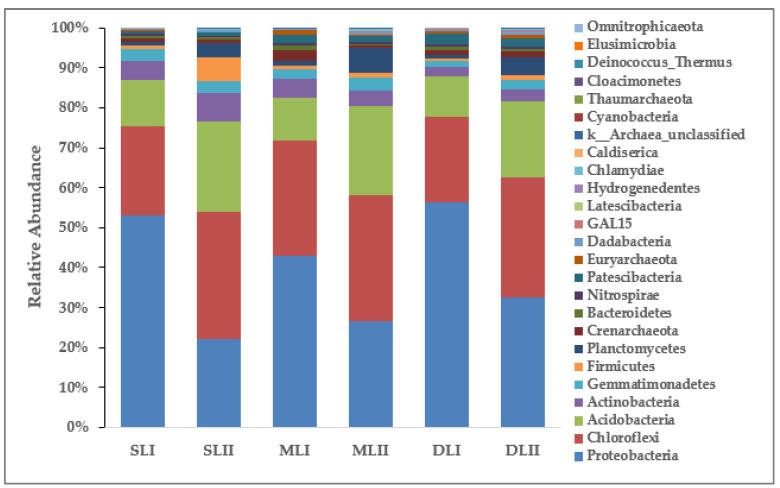
Relative abundance of the bacterial community composition at the phylum in different soil samples.

**Figure 7 ijerph-18-08661-f007:**
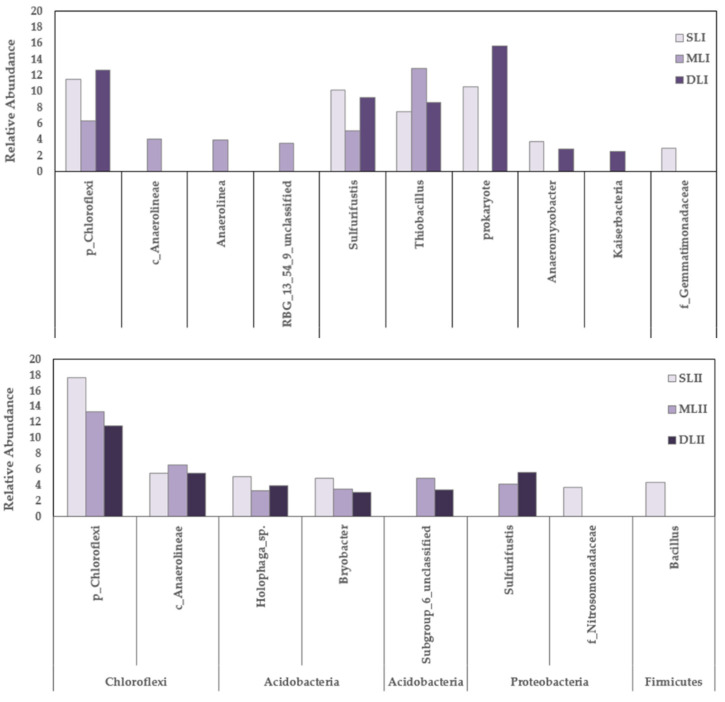
Relative abundance of the bacterial community composition at the genus level.

**Table 1 ijerph-18-08661-t001:** The primers used in this study.

No.	Primer Set	Size (bp)	Target	Reference(s)
1.	27F/1525R	1500	16S rRNA gene	[21]
2.	8F2/806R	802	16S rRNA gene	[22]
3.	fD1modF/16S1RR-B	568	16S rRNA gene	[23]

**Table 2 ijerph-18-08661-t002:** The heavy metal concentrations of all soil samples (metals in order of ion radius).

	mg/kg						
	Cr	Ni	Cu	Zn	Cd	Pb	pH
SL_I_	160.79 ± 6.36	133.78 ± 4.95	557.09 ± 13.76	603.06 ± 29.49	0.67 ± 0.12	68.03 ± 8.27	6.52 ± 0.01
ML_I_	165.99 ± 8.37	139.89 ± 11.49	528.74 ± 16.32	586.68 ± 25.92	0.69 ± 0.11	64.21 ± 6.89	6.70 ± 0.01
DL_I_	166.85 ± 6.30	136.95 ± 9.56	460.13 ± 15.92	544.80 ± 23.12	0.67 ± 0.06	59.08 ± 2.35	6.88 ± 0.01
SL_II_	153.53 ± 9.62	127.44 ± 12.70	583.12 ± 21.48	643.95 ± 24.26	0.68 ± 0.11	70.08 ± 9.78	6.09 ± 0.01
ML_II_	153.36 ± 9.11	123.91 ± 14.28	578.56 ± 21.77	594.21 ± 28.79	0.69 ± 0.08	69.15 ± 6.35	6.16 ± 0.01
DL_II_	163.25 ± 7.70	131.10 ± 15.14	588.88 ± 13.81	633.77 ± 16.60	0.84 ± 0.11	68.39 ± 5.95	6.23 ± 0.01
Variation in SL between two samplings	−4.5%	−4.7%	4.7%	6.8%	1.5%	3.0%	−6.6%
Variation in ML between two samplings	−7.6%	−11%	9.4%	1.3%	0.0%	7.7%	−8.1%
Variation in DL between two samplings	−2.2%	−4.3%	28%	16%	25%	16%	−9.4%
Monitoring value	109	112	280	363	<0.36	41.4	
Monitoring standard ^a^	175	130	120	260	2.5	300	
Control standard ^b^	250	200	200	600	5	500	

^a^: Soil monitoring standard values according to the soil pollution monitoring standard of the Taiwan Environmental Protection Agency. ^b^: Soil control standard values according to the soil pollution control standard of the Taiwan Environmental Protection Agency.

**Table 3 ijerph-18-08661-t003:** Characteristics of soil bacterial richness and diversity indices in different soil samples.

	OTUs	Observed_Species	Shannon	Simpson_Reciprocal	Chao1
SL_I_	33,130	353	6.01	25.12	353
ML_I_	39,622	331	6.27	32.63	331
DL_I_	30,014	275	5.46	17.38	275
SL_II_	21,681	264	5.72	20.50	264
ML_II_	18,085	323	6.15	29.22	323
DL_II_	15,635	309	6.21	33.77	309
Variation in DL between two samplings	−47.9%	12.8%	13.7%	94.3%	12.8%

**Table 4 ijerph-18-08661-t004:** Taxonomic identification of bacteria species with similarities on the NCBI Blast database. The tolerance index level of bacteria strains in copper/zinc enriched-media with different concentrations is presented.

LAB-ID	Closest NCBI xiaoshuangDatabase Match	Copper Enriched-Media Concentration (mg/L)					Zinc Enriched-Media Concentration (mg/L)				
		50	100	150	200	400	50	100	150	200	400
Cu_SL_I_-1	*Bacillus velezensis*	○	ν	-			○	○	ν	-	
Cu_ML_I_-1	*Bacillus albus*	ν	-				○	○	ν	-	
Cu_SL_II_-1	*Bacillus marisflavi*	○	○	○	○	-	○	ν	-	-	
Cu_SL_II_-2	*Bacillsu megaterium*	○	○	○	○	-	○	○	○	○	-
Cu_SL_II_-3	*Bacillus velezensis*	○	ν	-			○	○	ν	-	
Cu_SL_II_-4	*Bacillus marisflavi*	ν	-				○	ν	-	-	
Cu_ML_II_-1	*Bacillus amyloliquefaciens*	ν	-				○	○	ν	-	
Cu_ML_II_-2	*Bacillus aryabhattai*	○	○	○	○	-	○	○	○	ν	-
Cu_ML_II_-3	*Bacillus kribbensis*	ν	-				○	○	○	○	-
Cu_ML_II_-4	*Bacillus velezensis*	○	○	○	○	-	○	○	ν	-	
Cu_DL_II_-1	*Bacillus velezensis*	ν	-				○	○	ν	-	
Cu_DL_II_-3	*Bacillus marisflavi*	○	○	○	○	-	○	○	ν	ν	-
Zn_SL_II_-1	*Bacillus marisflavi*	ν	-				○	○	○	○	-
Zn_SL_II_-2	*Bacillus albus*	-					○	○	○	○	-
Zn_ML_II_-1	*Bacillus albus*	-					○	○	○	○	-
Zn_DL_II_-1	*Bacillus albus*	-					○	ν	ν	ν	-

Note 1: Cu (isolation and identification of copper-tolerant bacteria)_SL (surface layer)_I_ (first sampling)-1 (serial number); Zn (isolation and identification of zinc-tolerant bacteria)_SL (surface layer)_I_ (first sampling)-1 (serial number). Note 2: growing situation: ○ (grew well), ν (few microorganism growth), - (no growth).

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
