# Peer review of "Changes in the Composition of the Soil Bacterial Community in Heavy Metal-Contaminated Farmland"

_ijerph, 2021, doi:10.3390/ijerph18168661_

Round 1

Reviewer 1 Report

The topic of the manuscript is interesting and the authors had a good idea for a research project. The subject is relevant, aim, range and results were clearly defined and demonstrate a good scientific knowledge of the issues being discussed. The authors have presented the results of original research. I have no hesitation in recommending publication following minor revision.

General comments:

The subject fall within the general scope of the journal. The title clearly reflect the contents. Contribution is a new and original.

- Authors should further emphasize on the novelty of their work.

- Keywords:

Keywords should not be the same as words mentioned in the title

- Abstract:

The abstract Is sufficiently informative, especially when read in isolation.

- Introduction:

The introduction is well conducted since it addresses the issue and refers to relevant literature.

- Material and methods:

The methodology is well thought through. The description of materials and methods is informative to allow replication of the experiment.

- Results and Discussion:

The results are clearly represented. In my point of view this section is drawn up well.

-Conclusion 

The conclusion is rather concise but the approach is good. It can be replicated in future.

-References

The references are adequate and in agreement with the Guide for Authors.

Reviewer 2 Report

The presented manuscript by Tseng Shu-chun et al. is investigating the influence of heavy metals on the microbial communities within contaminated farmland soil. Methodologically, this work is adequate, and the presented results are interesting. However, some points, such as the discussion section, have to be improved. Some other critical points have to be addressed as well, as stated below in the comments.

Generally, this work needs some improvements to be on a good level for publication in this journal. English should be thoroughly checked. Also, there are a lot of misspellings throughout the text.

Comments:

Line 19. What does bioinformatics mean in this context? This should be re-phrased.

Chapter 2.3. Details on gel –electrophoresis and DNA concentration determination should be omitted, while additional information on sequence analysis should be added.

Line 163. There is no such thing as 16s rDNA, please change to 16s rRNA gene

Line 165. Please state explicitly which amplicon you used for the sequencing and what was its size, also the sequences should be deposited and accession numbers provided

Line 172. Why you have chosen to grow environmental bacteria at 37 C? This temperature probably leads to the selection of fast-growing bacteria, and this could be one of the reasons why you picked up only Bacillus as heavy-metal resistant bacteria. Additionally, in your OTU analysis, Firmicutes represent only a small fraction, so the growth conditions should be optimized to get more resistant isolates.

Table 1. Please provide standard deviations (you stated you have made measurements in triplicates), without SD, these results cannot be discussed at all. Why only deviation from DL is expressed?

Figure 3. The statistical significance for heavy metal changes should be calculated and specified in the bars.

Line 257. What does consistent changing trends mean? Are these changes statistically relevant? In further lines, the results from different time points are discussed and then a work regarding communities in different soil depths is cited. This is confusing and this paragraph should be re-written. Furthermore, in line 247 you were discussing how metal contamination is negatively correlated with bacterial diversity, and their results are quite contrary.  This should be thoroughly discussed as well.

Figure 6. Is there any literature describing the higher abundance of Acidobacteria in metal-contaminated soil?

Figure 6 & 7. It seems these are the same results presented with different charts?

Table 3. Unclear should be presented as a chart

Section 3.4. It is not clear whether the isolates were obtained only at 50 mg/L of heavy metals? In M&M 3 concentrations were stated (50, 100, 200 mg/L). Later these some of these bacteria were growing on higher metal concentrations. How is then possible no bacteria were isolated from higher metal concentration at first?

Tables 4,5 and 6 should be merged

Line 390 There is no proof that this is change is due to the change of metal composition. It might be, for example, because the pH has changed. This should be re-formulated.

Line 409. “Can be used” is a too strong statement based on the presented results

Reviewer 3 Report

This paper demonstrating the effect of metal pollution on the bacterial community is interesting. However, the presentation of the experiment and its results make it difficult to follow. The manuscript needs to be revised for better clarity.

More precisely:

  • line 17: for pH, put the decrease value in units and not %
  • line 18: "smaller ion radius" --> did you test ions with higher radius?
  • line 19: "bioinformatics at the three depths had changed" --> what does it mean?
  • line 20: what only Cu and Zn?
  • line 42: became
  • line 43: "cycle of toxicity" --> what does it mean?
  • lines 51 to 53: re-phrase, it is the microbial community that adapts, by changing its composition and structure.
  • lines 51 to 57: same idea, kind of repetition, re-structure (second part to put first).
  • line 62: protection of
  • line 63: define EPA
  • line 64: remove As.
  • line 66: what is control site?
  • line 81: the figure 1 shows location, not contamination (wrongly said)
  • lines 82-83: for which elements?
  • line 83: exceed
  • line 84: was (not is)
  • lines 85-86: "the grab sample method was chosen in this study" --> please be more specific
  • Figure 1: the text in the box is not readable, specify what is hotspot
  • lines 103-104: why air-dried then store at -80°C?
  • lines 117 to 119: remove at the beginning
  • lines 120-121: "should describe the sequence of primers and get citation" --> ???
  • lines 134 to 137: did you process data to remove chimera, too small fragments ... ? what was the method used for OTU clustering?
  • line 139: were
  • line 153: remove them and with
  • line 154: why only Cu and Zn?
  • line 163: same method of extraction as the soil ?
  • line 164: primers of which gene ?
  • lines 172 to 176: it seems a repetition with the first part... if I understand correctly, you grew isolated bacteria into agar solution with different Cu and Zn concentration and monitored their growth after 48h. Did you use these data to calculate µ and other parameters?
  • line 187 to 189: what about the other elements?
  • lines 195-196: pH has mainly an effect on metal mobility and availability, not total concentration...
  • lines 196 to 201: this is true but you performed aqua regia extraction, which gives total metal concentraiotn, not mobile fraction!
  • table 1: remind the meaning of SL, DL and ML. No statistical analysis to evaluate time effect?
  • figure 3: what is the purpose of this figure? The values for metal are the same than in table 1! Either keep this figure and add indication of monitoring value, standard and control standard, or remove it and add pH value in the table.
  • line 209 to 2019: goes with previous comment regarding pH and metal mobility
  • lines 229 to 231: please detail this in the M&M
  • lines 231 to 233: which database did you use to affiliate OTUs?
  • lines 233-234: don't understand
  • line 235: from 30 to 39000, it is high!
  • "significantly" --> it means that you performed statistical analysis? It is not shown anywhere!
  • table 2: you have 245505 sequences and in SL1 33130 OTUs which means only 8 reads for each sequence on average. It is not much !! Did you do replicates? Statistical analysis? In SL1, you 33130 OTUs and 350 species, so it means that many OTUs belong to the same species.
  • Figure 5: values not the same than in Table 2: for instance for DL1: 207+25+21+33 = 286 but 30014 in Table 2!
  • lines 256-257: not clear
  • lines 268-269: why not perform correlation analysis to confirm this?
  • line 272: remove "were" and "in this study"
  • line 277: changed in which way?
  • line 284: are (not is)
  • line 285: can be seen in Figure 6 also
  • line 289: advantageous???
  • lines 320-321: not clear
  • lines 322-323: not specified in the M&M
  • line 330: not specified in the M&M that an uncontaminated soil was studied
  • line 348: use the term "coding number" is not clear
  • lines 364-365: miss the end of the sentence
  • tables 5 and 6: specify the meaning of the circle and v.

Round 2

Reviewer 2 Report

The authors have taken into account most of the suggestions. The article is now well-written and easier to follow. Additionally, its scientific sound has been improved. 

Reviewer 3 Report

The authors performed the required modifications.

The article can now be accepted for publication.